# Comparison of two sentinel species *Gammarus fossarum* and *Dreissena polymorpha* for monitoring fecal viral contamination of continental waters

Marjorie Lortholarie,[1] Julie Do Nascimento,[1] Alain Geffard,[1] Isabelle Bonnard,[1] Christophe Gantzer,[2] Julie Challant,[2] Isabelle Bertrand,[2] Julie Loutreul,[3] Nicolas Boudaud,[3] Nicolas Delorme,[4] Arnaud Chaumot,[4] Olivier Geffard,[4] Mélissa Palos Ladeiro[1]

**ABSTRACT**   Biomonitoring is a widely used strategy for evaluating chemical contamination. Two species are used in particular in French rivers: *Dreissena polymorpha* and *Gammarus fossarum*. Although biomonitoring of fecal contamination has been initiated in the freshwater mussel *D. polymorpha*, relatively few studies are available for crustaceans. This study aimed to determine, for the first time, the value of the freshwater crustacean *G. fossarum* for monitoring viral contamination in comparison with *D. polymorpha*. Laboratory exposure was first performed to determine their ability to accumulate infectious F-specific RNA bacteriophages (FRNAPH). *In situ* exposure was then conducted to confirm this ability in the field, and also to study the bioaccumulation of viral genomes (FRNAPH and norovirus), along a *continuum* of fecal contamination. Laboratory exposure showed a significant difference in accumulation, with *D. polymorpha* (bioaccumulation factor [BAF] ~ 1) accumulating six times more infectious FRNAPH than *G. fossarum* (BAF ~ 0.3). *In situ* exposure confirmed that *G. fossarum* (BAF ~ 17) accumulated less infectious FRNAPH compared with mussels (BAF ~ 3,500). Moreover, FRNAPH and norovirus genomes were only quantified in mussels. Finally, the concentrations measured in *D. polymorpha* evidenced a fecal contamination gradient. *D. polymorpha* therefore appears to be more relevant than *G. fossarum* for monitoring viral contamination of fecal origin in freshwater environments.

**IMPORTANCE** Biomonitoring is a common method for assessing chemical contamination. In France, *Dreissena polymorpha* (zebra mussel) and *Gammarus fossarum* (a freshwater crustacean) are widely used species. This study aimed to compare, for the first time, their effectiveness in monitoring viral contamination, specifically F-specific RNA bacteriophages (FRNAPH) and noroviruses. Laboratory experiments showed that *D. polymorpha* accumulated six times more infectious FRNAPH than *G. fossarum*. Field exposures confirmed these findings, revealing that only mussels accumulated detectable levels of FRNAPH and norovirus genomes. Additionally, the viral concentrations in *D. polymorpha* reflected a gradient of fecal contamination across sites. These results demonstrate that *D. polymorpha* is significantly more effective than *G. fossarum* for monitoring viral contamination of fecal origin in freshwater environments, making it the more suitable species for such biomonitoring efforts.

**KEYWORDS**  crustacean, bivalve, biomonitoring, enteric viruses, F-specific RNA bacteriophages, norovirus

Address correspondence to Julie Do Nascimento, juliedonascimento@hotmail.fr.

The authors declare no conflict of interest.

Viral gastroenteritis, mainly caused by an enteric virus known as norovirus (NoV), is a major concern in terms of health problems associated with the microbiological contamination of water (1). Enteric viruses excreted by the population in wastewater can

contaminate the aquatic environment due to not being fully eliminated by wastewater treatment plants (WWTPs) before effluent is discharged into the natural environment (2). It is therefore crucial to monitor the occurrence of enteric viruses in aquatic environments. F-specific RNA bacteriophages (FRNAPH) are increasingly recommended as an indicator for monitoring NoV since they display similar behavior to enteric viruses, both in WWTP and in the environment (3, 4). In contrast to NoVs, which can only be routinely studied through their genome, FRNAPH can be studied through their genome, and also via their infectious particles. These different approaches can offer complementary information. The detection of infectious particles indicates recent contamination. However, the genome can shed light on the source of contamination as certain genogroups are more common in humans and others, in animals (5). These approaches may be relevant for monitoring enteric viruses, which are mainly of human origin, such as genogroups FRNAPH-II and NoV GII (6).

Measuring NoV and FRNAPH in aquatic environments is complex in terms of spatiotemporal variability and hydrodynamic properties (7). Dilution, adsorption to suspended particulate matter, sedimentation, and degradation over time all contribute to the low viral concentrations observed in water samples. Biomonitoring may be an appropriate solution for enteric viruses in aquatic environments, which is already the case for chemical contamination (8). In comparison to water, the biota can be an accumulative matrix, resulting in easier and more reliable measurement, while allowing integrated measurement over time and space. It can also be used to detect and quantify pollutants present at very low levels in the environment and, therefore, difficult to measure by directly testing water (9). As a result, numerous studies have been carried out to assess viral contamination in marine shellfish for human consumption (10) due to potential health risks. Although few studies have focused on freshwater environments, it may be worthwhile investigating whether the accumulation capacity observed in marine bivalves translates to the freshwater setting.

Invertebrates commonly used as sentinel species for biomonitoring European freshwater ecosystems include *Gammarus fossarum* and *Dreissena polymorpha* (8, 11). *D. polymorpha* is a freshwater bivalve used for biomonitoring due to its significant capacity to accumulate various pollutants, such as heavy metals and organic compounds, through filter feeding (12, 13). Several studies have effectively demonstrated their ability to accumulate microorganisms, including bacteria, protozoa, and viruses, both in the laboratory and *in situ* (14–17). However, mussels and gammarids have a different ecology (shredding vs filter) and could be complementary. *G. fossarum* is a freshwater amphipod crustacean known to be sensitive to and accumulate a wide range of chemical contaminants, such as metals, insecticides, polychlorinated biphenyls, and hydrocarbons (18, 19). In France, as part of the European Water Framework Directive, chemical status is partly monitored using bioaccumulation data from caged *G. fossarum* (Technical Note 2017), based on a standardized methodology for *in situ* exposure (AFNOR, XP T90-721). However, few studies have been carried out to assess the ability of *G. fossarum* to bioaccumulate human pathogenic microorganisms. To our knowledge, the work of Bigot-Clivot et al. (20) is the only evidence of its ability to accumulate protozoa. Numerous studies have demonstrated the value of *G. fossarum* and *D. polymorpha* for assessing environmental chemical contamination, thus confirming their potential interest in monitoring viral contamination.

This study aimed to determine the value of *G. fossarum* for monitoring viral contamination in freshwater in comparison with *D. polymorpha*. For this purpose, an initial approach was implemented in the laboratory setting to determine whether these two sentinel species were able to accumulate FRNAPH in the same manner. An active biomonitoring approach was then implemented during the second phase on a *continuum* of three sites downstream from WWTP discharge. As part of the latter approach, *D. polymorpha* and *G. fossarum* were caged and sampled at different times to determine their respective accumulation capacities for the different viral targets (total infectious FRNAPH, FRNAPH, and NoV genomes).

## MATERIALS AND METHODS

### Organisms

The organisms were collected from a reference site: Lac du Der for *D. polymorpha*, and a controlled farming area established on a former watercress farm for *G. fossarum*. Both species were then acclimatized under controlled laboratory conditions (dechlorinated tap water with the following physicochemical parameters: pH of 8 ± 0.5, conductivity of 464 ± 71 mS/cm, temperature of 13 ± 1°C, and oxygen saturation of 99.4 ± 4.3%) for at least 2 weeks and were fed twice-weekly *ad libitum*. *D. polymorpha* received a mix (50/50) of two commercial microorganism solutions (Nanno 3600 and Shellfish Diet, 1800; Planktovie, France), and *G. fossarum* were fed with alder (*Alnus glutinosa*) leaf discs.

The organisms were selected according to size. *G. fossarum* were selected using a certified sieve column (standard NF ISO 3310-1) and *D. polymorpha* by sampling organisms measuring 22 mm in length (11). *G. fossarum* underwent a second sorting according to sex, to select only males. The females were removed to avoid the biological variability associated with their reproductive cycle, marked by strong ovarian development, continuous embryonic growth in the marsupial pouch, and the release of juveniles at regular intervals depending on temperature (21, 22).

After acclimatization, three batches of 20 organisms (*G. fossarum*) or 20 digestive glands (*D. polymorpha*) were analyzed (see "Analyses" below) to check the absence of the viral targets investigated prior to exposure. Different methodological choices were made for the different organisms. Only the digestive gland was studied for mussels, in accordance with the literature (23). However, given the minimum mass required for analysis (1 g), and since dissection of the digestive cecum seemed impractical for large-scale applications, the whole organism was studied for *Gammarus*. The organisms were not rinsed before analysis, regardless of the type of exposure, whether in a laboratory setting or *in situ*.

### Experimental design

#### *Laboratory setting*

In the laboratory setting, 45 organisms from each species were divided into batches of 15, placed in three beakers under identical conditions (12°C and darkness), and exposed to infectious FRNAPH strain MS2 (genogroup I) (Fig. S1). The duration of exposure was 72 h, with renewal at two times (24 and 48 h). The exposure concentration was measured (see "Total infectious FRNAPH" below) after each renewal and averaged $1.1 \times 10^4$ for *G. fossarum* and $9.5 \times 10^3$ PFU/mL for *D. polymorpha*.

#### *In situ*

This study was conducted in the Vesle river (France) which receives effluent discharge from the Reims city WWTP (treatment capacity of approximately 470,000 population equivalents) between October and November 2022. The three chosen exposure sites corresponded to (i) the WWTP effluent discharge point (FC); (ii) a first site downstream from this discharge point, located 2.5 km down the river (MA); and (iii) a second site located 12 km down the river (CE). Several exposure durations were investigated: short (48 h, "D2"), medium (6 days, "D6"), and long (14 days, "D14") (Fig. S2).

Three cages, one for each sampling time, containing 65 mussels were placed at each site.

*G. fossarum* were caged in polypropylene cylinders with holes to allow free water circulation. The number of organisms placed depended on the duration of exposure and increased over time in anticipation of mortality, with a view to maintaining a pool of organisms for analysis (75, 125, and 150 organisms for 2, 6, and 14 days of exposure, respectively). Unlike mussels, which feed by filtering water, alder leaves were introduced as food support before and during exposure.

The buckets (*D. polymorpha*) and boxes (*G. fossarum*) were then dropped into the water column at a depth depending on the size of the aquatic structure (1.5 m max.).

Turbidity, oxygenation, and real conductivity were measured during caging (D0) and when the organisms were recovered (D2, D6, and D14) using a multi-parameter probe (Aqua TROLL 600). Field temperature probes were attached to the buckets to record temperature continuously during caging. Table S1 shows the mean values of physico-chemical parameters measured at the three sites.

## Analyses

### Sample preparation

After exposure, the three batches of 15 individuals (laboratory setting) and the three batches of 20 individuals (*in situ*) were divided into 15 mL tubes. Whole organisms were selected for *G. fossarum*, corresponding to a weight of 0.6 ± 0.2 and 0.58 ± 0.2 g, for the laboratory and *in situ* exposures, respectively. *D. polymorpha* were dissected to recover the digestive gland only (24). Sample weight was then 1.0 ± 0.2 and 0.77 ± 0.2 g, for the laboratory and *in situ* exposures, respectively. The samples were immersed in liquid nitrogen and stored at −80°C prior to analysis (see "Total infectious FRNAPH" and "FRNAPH and NoV genome detection and quantification" below).

On the day of analysis, the samples were thawed slowly on ice for 1 h. One volume of PBS and one volume of PBS-peptone 0.3% were then added to each sample. Grinding was performed on the Ultraturrax and, for the *in situ* exposure, an aliquot of 200 µL per sample was taken and stored at −80°C for the viral genome analysis (FRNAPH and NoV). The remaining homogenate was incubated on ice for 4 h and then centrifuged at 6,000 × *g* for 5 min to recover the supernatant for quantification of total infectious FRNAPH.

### Total infectious FRNAPH

Quantification of total infectious FRNAPH was carried out using the same method as previously described by Capizzi-Banas et al. (14) and Do Nascimento et al. (15) and based on the standard method ISO 10705-1 (25). The undiluted or 10-fold diluted superna-tants or water samples were added to refrigerated semi-solid agar supplemented with antibiotics together with host bacterium *Salmonella enterica* serovar Typhimurium WG49 (NCTC 12484). The resulting mixture was poured onto a petri dish and incubated at 37°C for 18 h. Lysis plate counts were used to calculate the concentration of total infectious FRNAPH in the sample in plaque-forming units per gram (PFU/g) or per milliliter (PFU/mL). The LOQ was defined as 1 PFU per g or per mL.

### FRNAPH and NoV genome detection and quantification

After thawing on ice, the organism samples were lysed with protease K and TRIzol. Chloroform was then added to recover the RNA. Following recommendations of the manufacturer, viral RNA was purified using the Pure Link RNA minikit (Thermo Fisher Scientific) (26). To summarize, 700 µL of supernatant was added to the same volume of ethanol, mixed, and transferred to the purification column with a collection tube. The column was centrifuged at 12,000 × *g* for 15 s. The RNA was washed with several successive buffer additions and centrifuged (15 s at 12,000 × *g*). The viral RNA was recovered in 40 µL of RNAse-free water. Enteric cytopathic bovine orphan virus (ECBO; ATCC VR-248) was added to each sample as a process control virus.

A volume of 100 mL of water per condition (for *in situ* exposure) was concentrated on a membrane (ZetaPlus 1 MDS). Lysis buffer (EasyMAG Lysis Buffer, bioMérieux) was then added to the filters for 10 min at room temperature to elute the nucleic acids. The samples were then purified using phenol-chloroform treatment (27), and magnetic silica beads (bioMérieux) were added to bind the RNA present in the sample. After several washing steps, the purified RNA samples were recovered in 70 µL.

NoV and FRNAPH genome detection was performed by RT-qPCR (Thermo Fisher). In this study, NoV GII and FRNAPH-II were detected in accordance with the

recommendations of NF EN ISO 15216-1 (2017) and the methodology described by Hartard et al. (28) and Wolf et al. (29). These two genogroups were studied, being the most commonly found at WWTP outlets due to their human (NoV GII) (30, 31) and predominantly human (FRNAPH-II) origin (5). As recommended by NF EN ISO 15216-1 (2017), samples were considered valid if ECBO extraction efficiency was greater than 1%. The number of RNA copies in each positive sample was estimated by comparing the Cq value with the standard curves. The final concentration was then adjusted based on the volume of nucleic acids analyzed and reported per gram of tissue (genome copies/g [gc/g]) or per milliliter of water (genome copies/mL [gc/mL]). The limits of quantification (LOQs) were 600 gc/g and 0.7 gc/mL for the organisms and water, respectively.

## Statistical analysis

The normality of the data were assessed using the Shapiro-Wilk test. The homogeneity of variances between the samples was also assessed using Levene's test. If the data followed a normal distribution and the variances were homogeneous ($P$ value > 0.05), parametric tests (ANOVA test or Student's $t$-test) were used. When at least one of the two conditions was not met ($P$ value < 0.05), non-parametric tests were used: Kruskal-Wallis test followed by a *post hoc* Dunn's multiple comparison test with the Bonferroni factor or Wilcoxon signed-rank test using RStudio (RStudio 1.4.1103).

Bioaccumulation factors (BAFs) were calculated using the concentrations measured in the organisms (PFU/g) at the end of exposure to concentrations above those measured in their exposure environments (PFU/mL).

## RESULTS AND DISCUSSION

### Laboratory exposure

The first exposure was designed to compare the infectious FRNAPH bioaccumulation levels for the two species under controlled laboratory conditions. At 0, 24, and 48 h, infectious phages were added to the beakers and, over the 72 h of exposure, the average concentration in the water was $1.1 \times 10^4$ ($n = 9$, SD = $8.1 \times 10^2$) and $9.5 \times 10^3$ ($n = 9$, SD = $6.5 \times 10^2$) PFU/mL, for *G. fossarum* and *D. polymorpha* exposure water, respectively. Figure S3 shows the mean concentrations of infectious FRNAPH measured in *D. polymorpha* and *G. fossarum* tissue after 72 h of exposure to similar concentrations. Infectious FRNAPH concentrations measured in mussel DG were significantly higher than those measured in *G. fossarum* ($P$ value < 0.01, Student's $t$-test), with averaged values of $1.8 \times 10^4$ ($n = 3$, SD = $2.9 \times 10^2$) and $3.4 \times 10^3$ ($n = 3$, SD = $8.1 \times 10^2$) PFU/g, respectively. These results reveal a difference in accumulation between the two organisms, which could be related to the characteristics of the organism itself. However, it could also be influenced by the different matrices used (digestive gland vs whole organism). As previously shown, *D. polymorpha* rapidly accumulates infectious FRNAPH under laboratory conditions, at high loads close to those tested here (14, 24). However, this study is the first to demonstrate the ability of *G. fossarum* to accumulate infectious FRNAPH in the laboratory.

To better compare the accumulation capacity of infectious FRNAPH between the two sentinel species, the BAFs were calculated and are shown in Fig. 1. Comparison of the BAF calculated in the laboratory setting showed that, despite the accumulation observed, the two species do not display significant concentrations relative to their environment (BAF of approximately 1). The concentrations measured in *G. fossarum* effectively appear to be three times lower than the average exposure load over 72 h (BAF of approximately 0.3). However, for *D. polymorpha*, the minimum BAF is 0.6 and the maximum is 2.2, indicating that the accumulated loads reflect exposure levels that are up to two times as high.

To our knowledge, only one other study has compared these two sentinel species in laboratory exposure by microbiological contamination. This study was carried out in the laboratory to investigate the accumulation of protozoa in gammarids (20) and touched on a comparison with *D. polymorpha* based on Palos Ladeiro et al. (32). The study

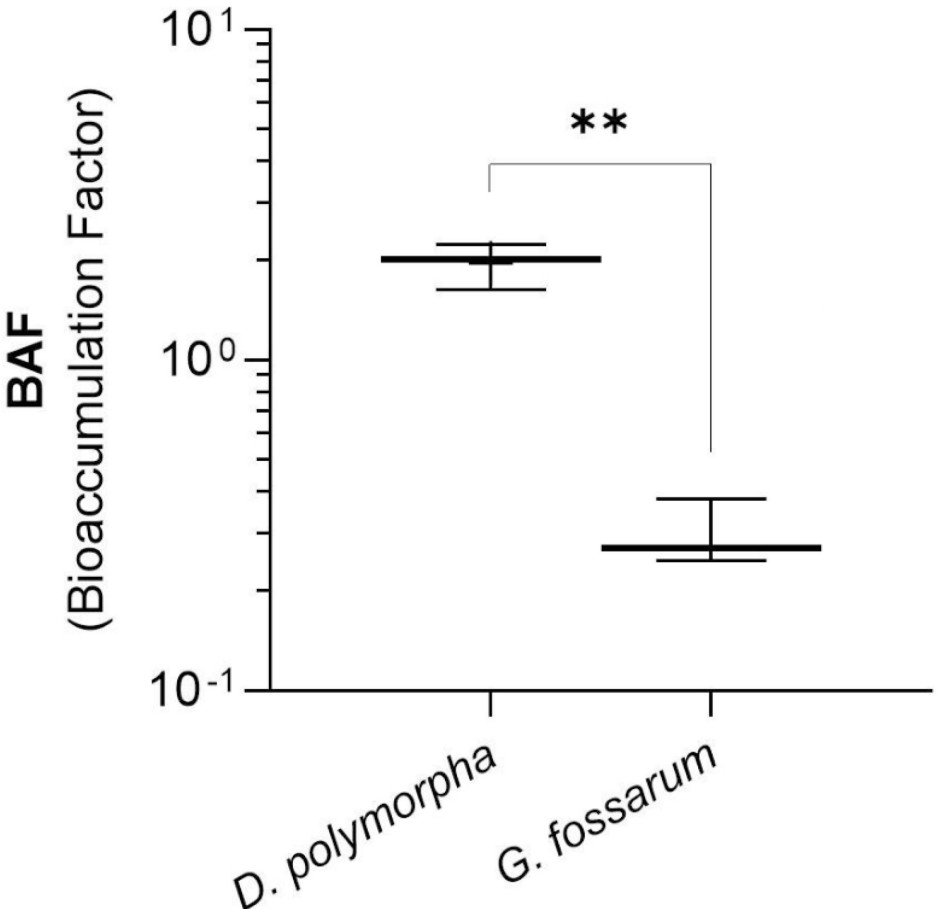

**FIG 1** Average bioaccumulation factor (BAF) values for total infectious FRNAPH by species (*D. polymorpha* and *G. fossarum*) in laboratory exposure ($n = 3$) Significant difference is labeled with asterisks (**, <0.01) according to the Wilcoxon test.

conducted by Bigot-Clivot et al. (20) showed that *G. fossarum* exposed to high concentrations were able to accumulate low concentrations of protozoa with a maximum BAF of 0.005 (at an exposure concentration of 200 oocysts per organism for 21 days). In comparison with mussels (32), the BAFs were of the same order of magnitude as those measured for *G. fossarum*. In our study, the BAF measured for infectious FRNAPH is of the same order of magnitude for *D. polymorpha* and *G. fossarum*. However, the BAF calculated in this study for phages, in both species, is 100 times higher than those determined for protozoa in the same species.

### *In situ* exposure

### *Levels of exposure of organisms to viral targets*

Figure S4 shows the average concentrations of viral targets (total infectious FRNAPH, FRNAPH-II, and NoV GII genomes) in water over the 14 days of exposure for each site. Whatever the target, significantly higher concentrations were observed at FC. The mean concentrations of total infectious FRNAPH in the water at FC were $1.3 \times 10^1$ PFU/mL ($n = 4$, SD = $5.3 \times 10^0$) compared with concentrations three times lower in the river downstream, averaging $4.0 \times 10^0$ ($n = 4$; SD = $1.3 \times 10^0$) and $3.0 \times 10^0$ ($n = 4$: SD = $3.1 \times 10^0$) PFU/mL for MA and CE, respectively. The same trend was observed for the FRNAPH-II genome, with average concentrations of $3.3 \times 10^2$ gc/mL ($n = 4$; SD = $5.9 \times 10^1$) in FC, $4.7 \times 10^1$ gc/mL ($n = 4$; SD = $4.2 \times 10^1$) in MA, and $8.3 \times 10^1$ gc/mL ($n = 4$; SD = $6.1 \times 10^1$) in CE. Although there was no significant difference for the NoV GII genome, it can be noted that the average concentrations at FC ($m = 2.8 \times 10^0$ gc/mL;

$n = 4$; SD = $2.1 \times 10^0$) are higher than at the two other sites MA ($m = 1 \times 10^{-1}$ gc/mL; $n = 4$; SD = $2 \times 10^{-1}$) and CE ($m = 2 \times 10^{-1}$ cg/mL; $n = 4$; SD = $2 \times 10^{-1}$). Therefore, the concentration of NoV GII tends to decrease along the *continuum*. These water analyses showed a difference in concentration between the treated effluent from the WWTP and the Vesle river, highlighting the potential impact of WWTP effluent on watercourses (2). However, these results do not allow us to identify an upstream-downstream gradient between the two sites in this river, as there is no significant difference between MA and CE.

In France, concentrations in the region of ND (not detected) to $3 \times 10^1$ PFU/mL, ND to $10^3$ gc/mL, and ND to $10^2$ gc/mL have been measured in WWTP effluent for total infectious FRNAPH, FRNAPH-II genome, and NoV GII genome, respectively (2, 5, 33). These concentrations are lower than those measured in treated wastewater, ranging from ND to $1.6 \times 10^1$ PFU/mL, ND to $2.3 \times 10^3$ gc/mL, and ND to $10^1$ cg/mL (2, 33) for total infectious FRNAPH, FRNAPH-II genome, and NoV GII genome, respectively. In the present study, the concentrations measured in WWTP effluent and in the river subject to these discharges are therefore of the same magnitude as those already observed in France.

## Spatiotemporal levels of viral targets in the organisms

The concentrations of the different viral targets measured in the two caged organisms at the three exposure sites (FC, MA, and CE) for 2, 6, and 14 days are shown in Fig. 2. For *G. fossarum*, only total infectious FRNAPH were quantified (ND to $2.2 \times 10^2$ PFU/g), as opposed to the viral genomes (FRNAPH-II and NoV GII). This difference between the detection of infectious particles and genome could be explained by higher sensitivity of the detection method applied for infectious FRNAPH than those used for genome detection, which are restricted to one genogroup. *D. polymorpha* accumulated total infectious FRNAPH ($1.8 \times 10^2$ to $4.9 \times 10^5$ PFU/g), FRNAPH-II genome ($6.8 \times 10^3$ to $2.6 \times 10^6$ gc/g), and NoV GII genome (ND to $1.1 \times 10^4$ gc/g) regardless of exposure site and duration. *G. fossarum* accumulated only total infectious FRNAPH during this exposure and only at one site, the most contaminated (FC), whereas *D. polymorpha* were able to accumulate all three viral targets (at all sites and all exposure times). While mussels exposed at FC accumulated total infectious FRNAPH consistently over time (no significant difference), *G. fossarum* accumulated total infectious FRNAPH only at the WWTP exit site (FC), only at short exposure times, and decreasing between D2 and D6. These concentrations effectively averaged $2.2 \times 10^2$ ($n = 3$; SD = $1.6 \times 10^2$) and $4.4 \times 10^1$ ($n = 3$; SD = $5.1 \times 10^1$) PFU/g for 2 and 6 days of exposure, respectively. In contrast, total infectious FRNAPH levels in *D. polymorpha* remained stable over time regardless of exposure site ($P$ value > 0.05). Finally, the concentrations measured in *G. fossarum* were significantly lower ($P$ value < 0.05) than those measured in *D. polymorpha* (e.g., 3 log difference at D6). Furthermore, the concentrations found in the organisms after *in situ* exposure relative to the water concentrations showed that the BAF (total infectious FRNAPH) at the FC site was 3,532 on average for *D. polymorpha* compared with 17.7 on average for *G. fossarum* (Fig. 3). The BAF calculated in this study shows that *G. fossarum* concentrates 429 times less the total infectious FRNAPH present in the environment than *D. polymorpha*. Although the study was not conducted on the same matrices, the level of differences in accumulation observed between *Gammarus* and *D. polymorpha* is such that it is easy to conclude that *Gammarus* appears to be a very low accumulator of these targets compared with mussels.

Three hypotheses could explain this difference in accumulation capacity. The methodologies used (for infectious and genomic detection) may be the first hypothesis for this difference between the two species. It is, effectively, important to note that we used whole gammarids for the analysis, as opposed to *D. polymorpha*, which can lead to sample dilution. To better understand these results, it would be worthwhile to extend the work on *G. fossarum* by studying the organotropism of these viral targets and, more specifically, by comparing the cecum with whole organisms. The second hypothesis is that *G. fossarum* is able to accumulate viral targets but displays faster and more efficient

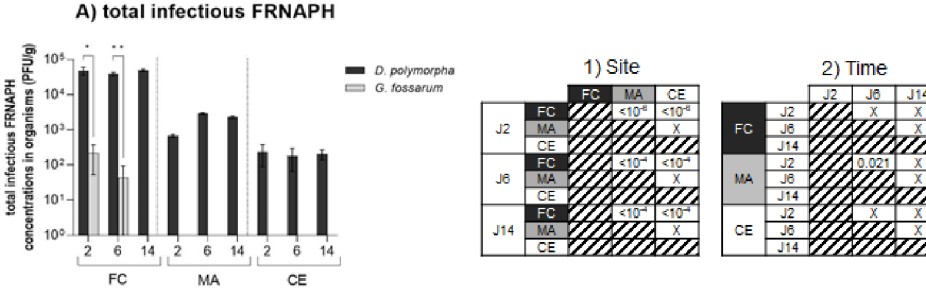

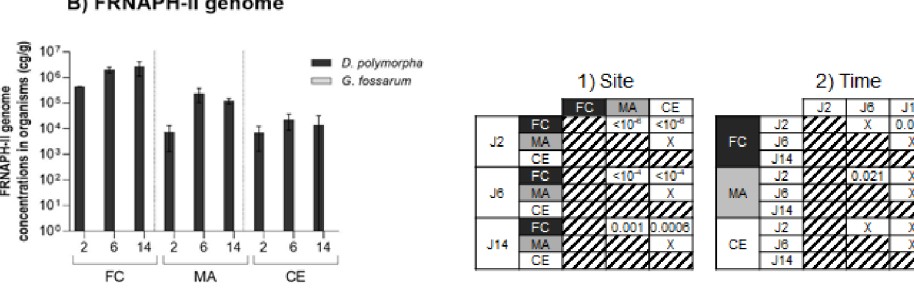

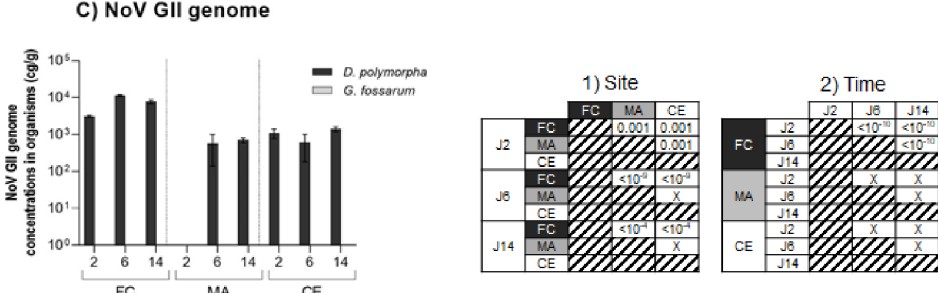

**FIG 2** Viral target concentrations: (A) total infectious FRNAPH (PFU/g) and (B) FRNAPH-II genome (cg/g) and (C) NoV GII genomes (cg/g) measured in *D. polymorpha* and *G. fossarum* samples at the three different exposure sites (FC, MA, and CE) according to exposure durations (2, 6, or 14 days). Error bars represent the mean with standard deviation (SD). Exposure sites are WWTP effluent (FC), 3.5 km (MA), and 12 km (CE) on the Vesle river, downstream of the WWTP. Exposure sites labeled with asterisks are significantly different (*, <0.05; **, <0.01) according to the Wilcoxon test. The *P* values determined according to the Kruskal-Wallis test comparing differences between sites and between exposure times for *D. polymorpha* data are given in tables (1 and 2) to the right of the graphs for each corresponding target. A non-significant difference (*P* value > 0.05) is indicated by the symbol "X" in the tables.

clearance compared with mussels. This hypothesis is, however, difficult to confirm. To our knowledge, little research has been conducted on gammarids immunity in response to microorganisms. Cornet et al. (34) highlighted the role of the proPO (prophenoloxidase) system in immune defenses in *G. pulex*. This system is generally known to play a role in protecting freshwater shrimp against bacteria (35), and may also, hypothetically, offer protection against viruses. According to the third hypothesis, the accumulation capacity of *G. fossarum* may be lower than that of mussels, for physiological and ecological reasons. One of the reasons for this may be a difference in feeding behavior. *D. polymorpha* is a bivalve able to filter a large volume of water (115–184 mL/organism/h) (36), and therefore has more contact with viral targets present in the water column and on particles compared with *G. fossarum*, a shredding organism. Another explanation could be related to the mode of exposure. The chambers were, effectively, exposed in the water

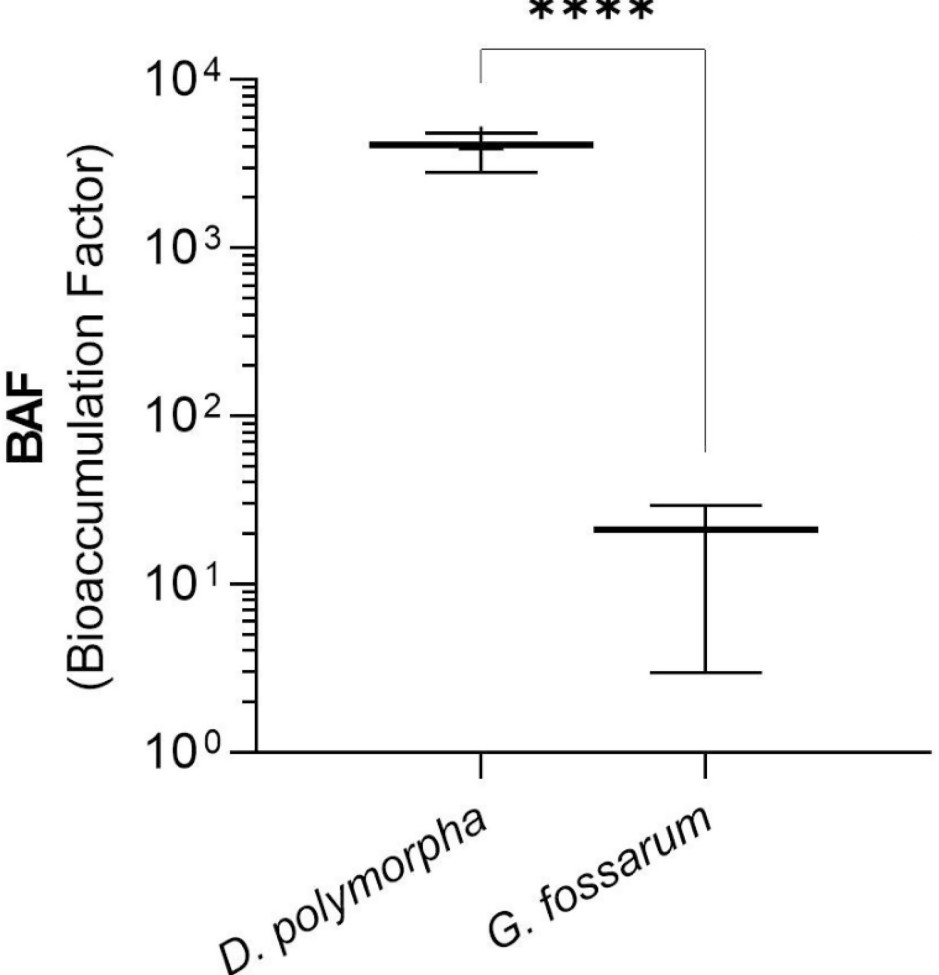

**FIG 3** Average bioaccumulation factor (BAF) values for total infectious FRNAPH by species (*D. polymorpha* and *G. fossarum*) in *in situ* exposure (FC site, *n* = 3 after 2 days. Significant difference is labeled with asterisks (****, <0.0001) according to the Wilcoxon test.

column, implying potentially lower exposure of *G. fossarum* and food material shredded by this benthic species. Studies have demonstrated the ability of enteric viruses to absorb particles suspended in water (37). It may therefore be more appropriate to attempt to use gammarids to monitor viral contamination at sediment level, as some viruses, including the infectious FRNAPH indicator, are capable of sedimentation (38–40). In addition, *D. polymorpha* survives better than gammarids, does not require feeding during exposure, and can be used in large streams and rivers. Moreover, the presence of other contaminants in the water, such as certain chemical contaminants, may have had a more detrimental effect on *G. fossarum* compared with *D. polymorpha*. Gammarids are known to be sensitive to numerous chemical contaminants (19) that may modulate their ability to accumulate viral loads. The exposure environment may, effectively, have a toxic effect, potentially influencing the accumulation of viral targets. However, in this study, the mortality rate for both species did not exceed 10% throughout exposure. This may be supported by the total infectious FRNAPH loads in *G. fossarum*, which decrease over time until they are no longer measured after 14 days.

These results suggest that *D. polymorpha* is more suitable than *G. fossarum* for monitoring viral contamination of continental waters, for both FRNAPH (infectious and genome) and NoVs. The use of *D. polymorpha* is effectively able to highlight spatiotemporal differences along a *continuum* for the three viral targets. However, some differences

were observed between the different exposure times. At the MA site, mussels exposed for two days had significantly lower concentrations of total infectious FRNAPH and FRNAPH-II genome than those exposed for 6 days. The same trend can be observed for the NoV genome, although this is not significant. These results could be related to the presence of embankment works at the MA site during exposure, which resulted in resuspension of particles. Mean turbidity at the MA site, located between FC and CE, is significantly higher and more variable ($m = 39.1 \pm 24.8$ NTU). At the start of exposure (D0) at the MA site, turbidity reached a value of 70 NTU, then decreased over time to 45 NTU after 2 days (D2), and to 14 NTU at the end of exposure (D14) (data not shown). This increased turbidity may have disturbed the mussels (41), resulting in less viral target accumulation. Significant differences can also be highlighted for viral genomes according to exposure time at the FC site. For the FRNAPH-II genome, the concentrations measured at D14 are higher than those measured at D2. For the NoV GII genome, differences in the concentrations measured in organisms exposed at all three times are observed, with the lowest concentrations found at D2 and the highest at D6. These variations at the FC site may be related to variations in WWTP effluent (2). Mussels exposed at sites downstream of WWTP effluent (MA and CE) appear to rapidly equilibrate with their environment within 48 h, since viral target concentrations do not differ significantly between exposure time steps. This observation has already been demonstrated for *D. polymorpha* exposed to infectious FRNAPH under controlled laboratory conditions (14, 24).

Spatial differences were observed in addition to the temporal differences described above. Mussels caged in the WWTP effluent (FC) had significantly higher concentrations of all three viral targets compared with mussels caged in the river downstream of the WWTP effluent (MA and CE). However, despite being nine kilometers apart, there were no differences in the organisms exposed in the Vesle river. Although there was no significant difference in FRNAPH concentrations (total infectious FRNAPH and FRNAPH-II genome), an upstream/downstream concentration gradient was observed. This may be due to concentration dilution and inactivation of infectious phages which increases with distance from the emission source (38), in this case WWTP discharge at FC. This study also shows that mussels exposed to environmental concentrations (MA and CE) within the range already measured in French rivers (2, 5, 33) were able to transcribe this contamination.

In addition to its spatio-temporal integration capacity, *D. polymorpha* is able to concentrate viral targets in the environment. Indeed, it was able to concentrate viral targets about 1,000-fold, with BAF of $1,321 \pm 12$, $937 \pm 61$, and $3,917 \pm 1,024$ for FRNAPH-II genome, NoV GII genome, and infectious FRNAPH, respectively. This ability to concentrate may be of particular interest in the case of low environmental viral loads which are difficult to isolate by spot water sampling without concentration methods.

This study aimed to compare two freshwater sentinel species, widely used for the assessment of chemical contamination, for biomonitoring fecal viral contamination. Their capacity for viral target accumulation was therefore investigated under controlled laboratory and field conditions in a river exposed to WWTP discharge. The accumulation of viral targets is higher in *D. polymorpha* compared with *G. fossarum* both in the laboratory and *in situ*. In addition, the mussels were able to translate spatiotemporal variation, which was not observed in gammarids. Although the two species do not display significant accumulation in the laboratory, the field experiment demonstrated a 1,000-fold increase in BAF for *D. polymorpha* compared with a 23-fold increase for *G. fossarum*. These differences between the two species may be explained by their ecology and ecophysiology, particularly since gammarids are shredders and mussels are filter feeders. Our results complement those from previous studies which highlighted the importance of *D. polymorpha* in comparison with other matrices, such as water or passive samplers (15).

## Synopsis

Comparison of two species demonstrates the usefulness of caged mussels for improved monitoring of enteric viruses in water.

## Highlights

- Difference in bioaccumulation of viral targets between the two sentinel species.
- *D. polymorpha* is more suitable than *G. fossarum* for monitoring viral contamination.
- Difference in bioaccumulation of viral targets between the two sentinel species.

## ACKNOWLEDGMENTS

This work was funded by the Research Program for Environmental and Occupational Health of Anses (2020/01/038). It was supported by the UMT Actia VIROcontrol (Joint Technological Unit), a joint partnership scheme between Actalia and LCPME, created and backed by the French Ministry of Food.

## AUTHOR AFFILIATIONS

[1]Université de Reims Champagne-Ardenne, Université Le Havre Normandie, INERIS, Normandie Univ, UMR-I 02 SEBIO, Reims, France
[2]Université de Lorraine, CNRS, LCPME, Nancy, France
[3]Food Safety Department, ACTALIA, Saint-Lô, Normandy, France
[4]INRAE, Laboratoire d'écotoxicologie Unité Riverly, Villeurbanne Cedex, France

## AUTHOR ORCIDs

Julie Do Nascimento http://orcid.org/0009-0002-5286-708X
Christophe Gantzer http://orcid.org/0000-0003-2287-4852
Isabelle Bertrand http://orcid.org/0000-0003-0743-4402
Nicolas Boudaud http://orcid.org/0000-0003-4031-4381

## AUTHOR CONTRIBUTIONS

Marjorie Lortholarie, Conceptualization, Data curation, Formal analysis, Investigation, Writing – original draft, Writing – review and editing | Julie Do Nascimento, Conceptualization, Data curation, Formal analysis, Investigation, Writing – original draft, Writing – review and editing | Alain Geffard, Funding acquisition, Project administration, Resources, Supervision, Validation | Isabelle Bonnard, Formal analysis | Julie Challant, Writing – review and editing | Isabelle Bertrand, Writing – review and editing | Julie Loutreul, Writing – review and editing | Nicolas Boudaud, Writing – review and editing | Nicolas Delorme, Writing – review and editing | Arnaud Chaumot, Methodology, Writing – review and editing | Olivier Geffard, Investigation, Methodology, Supervision, Validation, Writing – review and editing | Mélissa Palos Ladeiro, Conceptualization, Project administration, Resources, Supervision, Validation, Writing – review and editing.

## ADDITIONAL FILES

The following material is available online.

### Supplemental Material

**Supplemental figures and table (Spectrum01262-25-s0001.pdf).** Figures S1 to S4 and Table S1.

## Open Peer Review

**PEER REVIEW HISTORY (review-history.pdf).** An accounting of the reviewer comments and feedback.

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
