## [Reviewer comments · Microbiology Spectrum]

Microbiology Spectrum

Comparison of two sentinel species *Gammarus fossarum* and *Dreissena polymorpha* for monitoring fecal viral contamination of continental waters.

Marjorie Lortholarie, Julie Do Nascimento, Alain Geffard, Isabelle Bonnard, Christophe Gantzer, Julie Challant, Isabelle Bertrand, Julie Loutreul, Nicolas Boudaud, Nicolas Delorme, Arnaud Chaumot, Olivier Geffard, and Mélissa Palos Ladeiro

Corresponding Author(s): Julie Do Nascimento, Université de Reims Champagne-Ardenne

Review Timeline:

Submission Date:	April 22, 2025
Editorial Decision:	July 7, 2025
Revision Received:	September 22, 2025
Accepted:	November 3, 2025

Editor: Jeffrey Gardner

Reviewer(s): The reviewers have opted to remain anonymous.

Transaction Report:

DOI: <https://doi.org/10.1128/spectrum.01262-25>

Re: Spectrum01262-25 (**Comparison of two sentinel species *Gammarus fossarum* and *Dreissena polymorpha* for monitoring fecal viral contamination of continental waters.**)

Dear Dr. Julie Do Nascimento:

Thank you for the privilege of reviewing your work. Below you will find my comments, instructions from the Spectrum editorial office, and the reviewer comments.

Revision Guidelines

Sincerely,
Jeffrey Gardner
Editor
Microbiology Spectrum

Reviewer #1 (Comments for the Author):

The authors investigated the accumulation of F specific RNA bacteriophages (FRNAPH) and norovirus GII by *Gammarus fossarum* and *Dreissena polymorpha*. As for FRNAPH, both infectious particle and genome quantification were performed and the spatio-temporal variation of concentration was comparable in both methods (Figure 2). *D. polymorpha* presented higher accumulation efficiencies for FRNAPH and norovirus GII than *G. fossarum* and *D. polymorpha* can be used as a sentinel.

The technical concern of this study is an analytical method of viruses accumulated to *G. fossarum*. As whole body of *G. fossarum* was processed, viruses present on the surface of *G. fossarum* should be washed out properly and those taken into the body should be quantified. If the *G. fossarum* samples were not properly washed, the possibility of detecting viruses on their surface should be explained in the Results and Discussion.

English can be improved in places and professional English proofreading is recommended.

Specific comments

Line 2. Continental waters generally include waters from rivers, lakes, aquifers, glaciers but only river water receiving urban wastewater effluents was studied in this study. The title should be more specific to the contents.

Line 31-35. Quantitative description can be provided in the Abstract.

Line 31. It would be informative to explain *Dreissena polymorpha* is a freshwater mussel.

Line 33. The term "a fecal contamination gradient" is not used generally and this should be reworded.

Line 46. Appropriate references should be provided for this sentence.

Line 59. Appropriate references should be provided for FRNAPH.

Line 106. Please explain what the controlled laboratory conditions were like (i.e., source water, temperature, pH, etc.).

Line 165-170. When 15 individuals of *G. fossarum* were processed, weren't they washed before the addition of PBS and grinding? The average concentration of FRNAPH in the contaminated water was 1.1×10^4 PFU/mL and was higher than that in *G. fossarum* individuals (3.4×10^3 PFU/g) if 1 g was equivalent to 1 mL.

Line 237, 242. As for *G. fossarum*, the bioaccumulation factor (BAF) was less than 1 (Figure 1). The BAF was not calculated for genome copy numbers? It is possible that *G. fossarum* can take FRNAPH but it is doubtful whether this (BAF<1) can be called "accumulation". It is also possible that the contaminated water was retained on the surface of *G. fossarum* if they were not washed out properly before the analysis.

Line 273. $m = 2 \times 10^{-1}$ cg/mL

Line 334. Appropriate references should be provided not only for FRNAPH, but also for enteric viruses (Hassard et al. 2016) including norovirus (Miura et al. 2011).

Hassard F, Gwyther CL, Farkas K, Andrews A, Jones V, Cox B, Brett H, Jones DL, McDonald JE, Malham SK. Abundance and Distribution of Enteric Bacteria and Viruses in Coastal and Estuarine Sediments-a Review. *Front Microbiol.* 2016;7:1692. doi: 10.3389/fmicb.2016.01692.

Miura T, Masago Y, Sano D, Omura T. Development of an effective method for recovery of viral genomic RNA from environmental silty sediments for quantitative molecular detection. *Appl Environ Microbiol.* 2011;77(12):3975-81. doi: 10.1128/AEM.02692-10.

Reviewer #1 (Comments for the Author):

The authors investigated the accumulation of F specific RNA bacteriophages (FRNAPH) and norovirus GII by *Gammarus fossarum* and *Dreissena polymorpha*. As for FRNAPH, both infectious particle and genome quantification were performed and the spatio-temporal variation of concentration was comparable in both methods (Figure 2). *D. polymorpha* presented higher accumulation efficiencies for FRNAPH and norovirus GII than *G. fossarum* and *D. polymorpha* can be used as a sentinel.

The technical concern of this study is an analytical method of viruses accumulated to *G. fossarum*. As whole body of *G. fossarum* was processed, viruses present on the surface of *G. fossarum* should be washed out properly and those taken into the body should be quantified. If the *G. fossarum* samples were not properly washed, the possibility of detecting viruses on their surface should be explained in the Results and Discussion.

Thank you for your comment, the hypothesis seems very pertinent and could have explained the overestimation of concentrations in *G. fossarum*. However, here, even if there is the presence of viruses absorbed and adsorbed by *G. fossarum*, we observe that it accumulates very little, if any, of the viral targets. However, this does not change the conclusions of comparisons between the two species.

English can be improved in places and professional English proofreading is recommended.

Specific comments

Line 2. Continental waters generally include waters from rivers, lakes, aquifers, glaciers but only river water receiving urban wastewater effluents was studied in this study. The title should be more specific to the contents.

Thank you for your comment, we understand your point of view. However, these viral targets are not only released into the environment via WWTP discharges into rivers, but can also be discharged via leaching from agricultural areas with livestock farms and areas with WWTP sludge spreading, and thus also end up in lakes.

Line 31-35. Quantitative description can be provided in the Abstract.

Thank you very much, indeed the abstract appears more complete. We have therefore added the average BAF values under laboratory and in situ conditions for both species (see lines 30-32).

Line 31. It would be informative to explain *Dreissena polymorpha* is a freshwater mussel.

Thank you very much for your comment, which we have taken into account (see line 31).

Line 33. The term "a fecal contamination gradient" is not used generally and this should be reworded.

We have reworded the term (see line 33), thank you for your comment.

Line 46. Appropriate references should be provided for this sentence.

Following your comment, a reference has been added to support our claims (see line 46).

Line 59. Appropriate references should be provided for FRNAPH.

Thank you, we have added the reference as requested (see line 60).

Line 106. Please explain what the controlled laboratory conditions were like (i.e., source water, temperature, pH, etc.).

Thank you for your comment. We have detailed the controlled laboratory conditions for the organism acclimation phase (see line 104-106).

Line 165-170. When 15 individuals of *G. fossarum* were processed, weren't they washed before the addition of PBS and grinding? The average concentration of FRNAPH in the contaminated water was 1.1×10^4 PFU/mL and was higher than that in *G. fossarum* individuals (3.4×10^3 PFU/g) if 1 g was equivalent to 1 mL.

Thank you for raising this question, indeed, *G. fossarum* were not rinsed before adding the PBS and grinding, as added in material and methods sections (see lines 121-122). In general, in laboratory exposure, concentrations of infectious FRNAPH in water are higher than in organisms, including *D. polymorpha* (Capizzi-Banas et al., 2021; Do Nascimento et al., 2024), which explains the BAF below or around 1 in these studies but also in the present study.

Line 237, 242. As for *G. fossarum*, the bioaccumulation factor (BAF) was less than 1 (Figure 1). The BAF was not calculated for genome copy numbers? It is possible that *G. fossarum* can take FRNAPH but it is doubtful whether this (BAF<1) can be called "accumulation". It is also possible that the contaminated water was retained on the surface of *G. fossarum* if they were not washed out properly before the analysis.

The aim of the laboratory exposure was to compare levels of infectious FRNAPH accumulation, not FRNAPH genome, in the two species *G. fossarum* and *D. polymorpha*. To reduce confusion for the reader, the following sentence has been reasserted in the results section (lines 234-237) : « However, these results should be treated with caution, as the organisms were not rinsed prior to analysis. This could mean that the viral target was adsorbed onto the surface of *G. fossarum*, which would lead to an overestimation of accumulated concentrations ».

Line 273. $m=2 \times 10^{-1}$ cg/'m'L

Thanks for pointing out this typo, we have corrected it (line 267).

Line 334. Appropriate references should be provided not only for FRNAPH, but also for enteric viruses (Hassard et al. 2016) including norovirus (Miura et al. 2011).

Thank you for sending us these references, which we have read and added to our text (line 327).

Hassard F, Gwyther CL, Farkas K, Andrews A, Jones V, Cox B, Brett H, Jones DL, McDonald JE, Malham SK. Abundance and Distribution of Enteric Bacteria and Viruses in Coastal and Estuarine Sediments-a Review. *Front Microbiol.* 2016;7:1692. doi: 10.3389/fmicb.2016.01692.

Miura T, Masago Y, Sano D, Omura T. Development of an effective method for recovery of viral genomic RNA from environmental silty sediments for quantitative molecular detection. *Appl Environ Microbiol.* 2011;77(12):3975-81. doi: 10.1128/AEM.02692-10.

Re: Spectrum01262-25R1 (**Comparison of two sentinel species *Gammarus fossarum* and *Dreissena polymorpha* for monitoring fecal viral contamination of continental waters.**)

Dear Dr. Julie Do Nascimento:

Your manuscript has been accepted, however I ask that you consider the comments of Reviewer #1 when you proof your manuscript.

I am forwarding it to the ASM production staff for publication. Your paper will first be checked to make sure all elements meet the technical requirements. ASM staff will contact you if anything needs to be revised before copyediting and production can begin. Otherwise, you will be notified when your proofs are ready to be viewed.

Sincerely,
Jeffrey Gardner
Editor
Microbiology Spectrum

Reviewer #1 (Comments for the Author):

Thank you for addressing the comments from the Reviewer. There are some minor comments that can be addressed to improve the quality of manuscript.

Specific comments

Reviewer #1. Line 2. Continental waters generally include waters from rivers, lakes, aquifers, glaciers but only river water receiving urban wastewater effluents was studied in this study. The title should be more specific to the contents.

Authors. Thank you for your comment, we understand your point of view. However, these viral targets are not only released into the environment via WWTP discharges into rivers, but can also be discharged via leaching from agricultural areas with livestock farms and areas with WWTP sludge spreading, and thus also end up in lakes.

Reviewer #1. I agree that FRNAPH and norovirus GII are released into the environment and lake waters can be contaminated with enteric viruses. However, *Gammarus fossarum* and *Dreissena polymorpha* were not compared for monitoring the fecal viral contamination in lakes, aquifers, and glaciers in this study. I don't think *D. polymorpha* can be used as a sentinel in aquifers or glaciers. The title will be more informative if you could consider more appropriate terms as a research article.

Reviewer #1. Line 32. Please spell out and then abbreviate "BAF".

Reviewer #1. Line 237, 242. As for *G. fossarum*, the bioaccumulation factor (BAF) was less than 1 (Figure 1). The BAF was not calculated for genome copy numbers? It is possible that *G. fossarum* can take FRNAPH but it is doubtful whether this (BAF<1)

can be called "accumulation". It is also possible that the contaminated water was retained on the surface of *G. fossarum* if they were not washed out properly before the analysis.

Authors. The aim of the laboratory exposure was to compare levels of infectious FRNAPH accumulation, not FRNAPH genome, in the two species *G. fossarum* and *D. polymorpha*. To reduce confusion for the reader, the following sentence has been reasserted in the results section (lines 234-237) : « However, these results should be treated with caution, as the organisms were not rinsed prior to analysis. This could mean that the viral target was adsorbed onto the surface of *G. fossarum*, which would lead to an overestimation of accumulated concentrations ».

Reviewer #1. The sentence has NOT been reasserted in the results section of submitted manuscript (Spectrum01262-25R1-Merged_PDF).

Reviewer #2 (Comments for the Author):

I am happy with the responses provided by the authors and the revision made.